# Parameters of Off-Vertical Axis Rotation in Unilateral and Bilateral Vestibulopathy and Their Correlation with Vestibular Evoked Myogenic Potentials

**DOI:** 10.3390/jcm10040756

**Published:** 2021-02-13

**Authors:** Shin Hye Kim, Sang-Yeon Lee, Ji-Soo Kim, Ja-Won Koo

**Affiliations:** 1Uijeongbu Eulji Medical Center, Department of Otorhinolaryngology-Head and Neck Surgery, Eulji University College of Medicine, Uijeongbu 11759, Korea; shinhye1225@gmail.com; 2Department of Otorhinolaryngology-Head and Neck Surgery, Seoul National University Bundang Hospital, Seoul National University College of Medicine, Seongnam 13620, Korea; maru4843@hanmail.net; 3Department of Neurology, Seoul National University Bundang Hospital, Seoul National University College of Medicine, Seongnam 13620, Korea; jisookim@snu.ac.kr

**Keywords:** off-vertical axis rotation, vestibular function, otolith function, otolith, vestibular evoked myogenic potential

## Abstract

Off-vertical axis rotation (OVAR) is a laboratory test to assess the otolith function. This study aimed to analyze the parameters of OVAR in patients with unilateral vestibular hypofunction (UVH) and bilateral vestibulopathy (BVP), and to correlate the parameters of OVAR with those of VEMPs. Ten healthy volunteers, 41 UVH, and 13 BVP patients performed OVAR. Bias component (BIC) and modulation component (MOC) of UVH and BVP patients were compared with those of healthy controls. BIC and MOC were correlated with amplitude and interaural difference (IAD) of cervical VEMP (cVEMP) and ocular VEMP (oVEMP). In UVH patients, the direction of BICs to affected side rotation were reversed and the absolute value of BICs were decreased when they were compared to healthy controls. In BVP patients, BICs were markedly attenuated. MOCs were not changed in UVH and BVP patients. There was no statistically significant correlation between VEMPs and OVAR.

## 1. Introduction

Clinical tests have been introduced for objective evaluation of otolith function, such as subjective visual vertical, fundus photography, and vestibular evoked myogenic potential (VEMP). In addition, eccentric rotation and off-vertical axis rotation (OVAR) have been used limitedly in the research laboratories. Eccentric rotation stimulates the otolith organ in a manner similar to a centrifuge: head rotation in the vertical direction is oriented away from the center of the rotation axis. During eccentric rotation, the angular vestibular-ocular reflex (VOR) occurs due to semicircular canal stimulation [1]. The linear VOR also occurs because of linear acceleration during off-axis side rotation [2].

During the OVAR test, the rotation axis is tilted away from the earth vertical and bilateral otolith organs, and there is concurrent semicircular canal stimulation [3]. When patients are positioned with their noses upward, downward, or rotated around the axis (yaw rotation) in clockwise (CW) and counter-clockwise (CCW) directions, these stimuli change the otolith response to gravity, along with the semicircular canal response [4]. Ocular reflexes to OVAR include the semicircular canal–ocular reflex, otolith–ocular reflex, and semicircular canal–otolith interaction [5].

The OVAR test involves otolith stimulation through changes in the direction of gravity acceleration in patients who are seated on a rotating chair tilted forward or backward. During trapezoid velocity yaw rotation in the off-vertical axis, the VOR generated by the horizontal semicircular canal decreases exponentially, although the VOR generated by the otolith organ is preserved as the direction of gravity changes. Therefore, the OVAR test may provide useful information concerning the functions of otolith organs, as well as interactions between otolith organs and semicircular canals.

The VEMP is now widely used as a simple and convenient test to evaluate the integrity of the vestibular afferent pathway, which originates from the otolith end organs [6]. The cervical vestibular evoked myogenic potential (cVEMP) is a sternocleidomastoid muscle (SCM) inhibitory response generated by auditory stimulation of the saccule [7], and the ocular vestibular evoked myogenic potential (oVEMP) is an inferior oblique muscle excitatory response generated by stimulation of the utricle [8]. Although both VEMP and OVAR tests are useful for evaluating otolith function, distinct stimuli are used (i.e., auditory stimuli for the VEMP test vs. rotation stimuli for the OVAR test) and different types of responses are measured (i.e., myogenic potentials for the VEMP test vs. eye movements for the OVAR test). To our knowledge, no studies have compared the VEMP test and OVAR tests.

This study was conducted to analyze OVAR parameters in patients with unilateral vestibular hypofunction (UVH) or bilateral vestibulopathy (BVP), and to assess their relationships with VEMP parameters.

## 2. Materials and Methods

### 2.1. Participants

In total, 10 healthy volunteers, 41 patients with UVH, and 13 patients with BVP were included in this study. The 10 healthy volunteers (mean age: 31.9 ± 13.0 years; 5 males) had no history of dizziness and showed normal vestibular function in both caloric test and rotation chair test.

The 41 patients with UVH (mean age: 50.9 ± 12.3 years; 17 males) had no spontaneous nystagmus and showed >25% canal paresis on the caloric test. Strict diagnostic criteria for BVP were a sum of the peak slow phase velocities of the right warm (RW), and right cold (RC), left warm (LW), and left cold (LC) <12°/s (<6 sum of each side) on the caloric test, and reduced gain and increased phase lead on the rotation chair test. In this study, 13 patients were included in the BVP group (mean age: 48.2 ± 14.9 years; 7 males). Of these 13 patients, 9 met the criteria for BVP. Although the remaining four patients (patients 10–13) did not meet the strict criteria for BVP, they were analyzed in BVP group to see responses of subjects in the spectrum between UVH and BVP since their caloric responses were substantially low.

### 2.2. OVAR

The OVAR test (I-Portal^®^ Neuro-Otologic Test Center; Neuro Kinetics Inc., Pittsburgh, PA, USA) was performed in all participants in this study. Following step velocity stimulation in the earth vertical axis (40 s; 60°/s), the chair was tilted to 20° after disappearance of the horizontal VOR. The tilt position was maintained for six cycles with CW or CCW rotations.

Eye movements were recorded using a binocular video oculography system (I-Portal^®^; Neuro Kinetics Inc., Pittsburgh, PA, USA). Horizontal, vertical, and torsional eye position data were digitally differentiated and the fast components of nystagmus identified and removed by the system software with operator intervention. We analyzed the slow phase velocity of the horizontal component of nystagmus during the OVAR based on sine waves with a baseline over 0°/s (Figure 1A). The center of offset is regarded as the bias component (BIC), and the amplitude of the sine wave above or below the offset is regarded as the modulation component (MOC) (Figure 1B). BICs and MOCs were measured with both CW and CCW rotation.

In left UVH, the direction of rotation of the affected side is CCW, and that of the healthy side is CW. In the analysis of UVH patients, all the affected side was assumed in the right ear. Based on this assumption, in left UVH patients, BICs of affected side rotation (CCW rotation) were analyzed as CW with opposite value, and heathy side rotation (CW rotation) were as CCW with opposite value.

### 2.3. VEMP

Air-conduction cVEMP tests using 500 Hz tone burst stimuli with an intensity of 105 dB nHL through calibrated headphone, and bone-conduction oVEMP tests using tapping stimuli with the peak amplitude of 85 dB FL were performed in all patients with UVH or BVP to identify correlations with the OVAR results. Each test has been described in detail previously [9,10]. In brief, for the cVEMP test, we placed the subjects on a bed in a supine position, raised their head to approximately 30° from the horizontal, and rotated it contralaterally. The active, reference, and ground electrodes were placed at the midpoint of the SCM, medial clavicle, and center of the forehead, respectively. The amplified EMG activities of the SCM were also monitored and digitized at 1 kHz using an analog-to-digital converter (NI PXl-4461; National Instruments, Austin, TX, USA). The LabVIEW program (National Instruments) was used to analyze the peak-to-peak amplitudes and calculate the mean tonic activation during the recording. The absolute cVEMP amplitude was then normalized against the mean tonic activation of the SCM during the recording. For the oVEMP test, we placed the subjects on a bed in a supine position with the head supported on a pillow. In each eye, the active recording electrode was placed on the infra-orbital ridge 1 cm below the center of each lower eyelid and the reference electrode was placed about 2 cm below that active electrode. The ground electrode was placed on the forehead. During oVEMP recording, the subjects had an upward gaze approximately 25° above straight ahead and maintained a small fixation point approximately 60 cm from the eyes. The signals were amplified by differential amplifiers (bandwidth 10–2000 Hz), and the unrectified signals were averaged (*n* = 100). The normalized amplitude (N-amplitude) of cVEMP and n10–p15 amplitude (μV) of oVEMP were measured. The normal ranges of the N-amplitude of cVEMP and n10–p15 amplitude of oVEMP were defined as 2.26–9.78 and 4.26–56.46 μV, respectively. As an index of the difference between ears, interaural differences (IADs, where IAD = 100(%) × (Amp[left] − Amp[right])/(Amp[left] + Amp[right])) in VEMPs were also compared with the affected side BICs and MOCs of OVAR. The normal ranges of IAD were defined as −22.5% to 22.5% for the cVEMP and −21.5% to 21.5% for the oVEMP [10].

To determine the relationship between VEMP and OVAR, the affected side cVEMP and oVEMP amplitudes were compared with the affected side BICs and MOCs of OVAR. To analyze the correlations between VEMP and OVAR parameters, the OVAR BIC values were directly compared with the amplitudes and IADs of VEMP.

### 2.4. Statistical Analysis

Data analyses were conducted using IBM SPSS Statistics for Windows software (version 21.0; IBM Corp., Armonk, NY, USA). ANOVAs were used to compare the means of three variables. For bivariate comparison, the Mann–Whitney U test was used for continuous variables. To compare the values of OVAR and VEMP directly, Pearson correlation analysis was used. All reported *p*-values were two-sided, and *p* < 0.05 was considered statistically significant.

## 3. Results

### 3.1. OVAR Results in Normal Volunteers 

For the 10 normal volunteers, the mean ± standard deviation (SD) BIC values for CW and CCW rotation were −3.10 ± 1.70°/s and 2.92 ± 2.46°/s, respectively (Appendix A). The mean absolute BIC value was 3.01 ± 2.06°/s. The sum of the absolute BIC values (|CW| + |CCW|) was 6.02 ± 3.72°/s. The mean ± SD MOC values for CW and CCW rotation were 3.25 ± 1.49° and 3.36 ± 1.29°, respectively. Regardless of rotation direction, the mean absolute MOC value was 3.31 ± 1.36°.

### 3.2. OVAR Results in Patients with UVH

The BIC and MOC values of the 41 patients with UVH are shown in Appendix A. For the right UVH patient, the mean ± SD BIC values for CW and CCW rotation were −0.53 ± 1.33°/s and 1.65 ± 1.38°/s, respectively. The sum of the absolute BIC values (|CW| + |CCW|) was 2.80 ± 1.82°/s. The mean ± SD MOC values for CW and CCW rotation were 4.15 ± 2.15° and 3.99 ± 2.64°, respectively. The mean absolute MOC value was 4.07 ± 2.38°.

For the left UVH patients, the mean ± SD BIC values for CW and CCW rotation were −2.59 ± 2.07°/s and −1.46 ± 2.25°/s, respectively. The sum of the absolute BIC values (|CW| + |CCW|) was 4.62 ± 3.50°/s. The mean ± SD MOC values for CW and CCW rotation were 3.77 ± 2.60° and 3.28 ± 1.99°, respectively. The mean absolute MOC value was 3.52 ± 2.30°.

In the analysis assuming that all affected sides were right ears, the mean ± SD BIC values for CW rotation (affected side) and CCW rotation (healthy side) were 1.10 ± 1.90°/s and 2.13 ± 1.81°/s, respectively. The MOC values for affected and healthy side rotation were 3.70 ± 2.04°/s and 3.80 ± 2.55°/s, respectively.

### 3.3. OVAR Results in Patients with BVP

In the 13 patients with BVP, the mean ± SD BIC values for CW and CCW rotation were −0.10 ± 0.93°/s and 0.03 ± 0.77°/s, respectively. The sum of the absolute BIC values (|CW| + |CCW|) was 1.30 ± 0.85°/s. The mean ± SD MOC values for CW and CCW rotation were 3.32 ± 1.27° and 2.21 ± 1.29°, respectively (Appendix A).

### 3.4. Comparison of OVAR Results among Groups 

Although the BIC values obtained during CW and CCW rotation in normal volunteers were symmetric with opposite signs, those in patients with UVH were asymmetrical and lateralized to the unaffected side rotation (i.e., CCW in right UVH patients and CW in left UVH patients) (Figure 2A). To simplify the comparison of BICs between patients with UVH and normal volunteers, all affected sides were assumed to be right ears. The mean BIC values during affected side rotation (CW in right UVH patients) and unaffected side rotation (CCW in right UVH patients) were similar but with opposite signs, indicative of directional preponderance (Figure 2B). In patients with BVP, the BIC values approached 0 and varied depending on the remaining vestibular function. The sum of the absolute value of BICs (|CW| + |CCW|) differed significantly among groups (Figure 2C). In contrast, the MOCs did not differ significantly among groups, regardless of the rotation direction. Notably, the MOCs did not differ even among the patients with BVP (Figure 3A,B).

### 3.5. Relationships between OVAR and VEMP in Patients with UVH

To determine the relationships between OVAR and VEMP, the OVAR results were compared with the cVEMP/oVEMP results in 41 patients with UVH using the Mann–Whitney U test. The patients were divided into normal and abnormal groups according to the affected side VEMP amplitude, and the affected side bias and MOCs of OVAR were compared between the two groups. The BIC and MOCs of OVAR were compared between the normal cVEMP N-amplitude and abnormal cVEMP N-amplitude groups, and no significant differences were seen (Figure 4). The BIC and MOC of OVAR were also compared between the normal n10–p15 amplitude and abnormal n10–p15 amplitude oVEMP groups, which showed no significant difference (Figure 4).

The correlations between VEMP and OVAR parameters in patients with UVH and BVP were examined to further characterize the relationships between VEMP and OVAR (Figure 5). However, no significant correlations were observed between the IADs of cVEMP and BICs with CW/CCW OVAR (*r* = −0.204, *p* = 0.201 and *r* = −0.056, *p* = 0.730, respectively). In addition, no significant correlations were observed between the IADs of oVEMP and BICs with CW/CCW OVAR (*r* = 0.121, *p* = 0.452 and *r* = 0.047, *p* = 0.773, respectively).

### 3.6. Relationships between OVAR and VEMP in Patients with BVP 

Responses in the cVEMP (7/13, 53.8%) and oVEMP (5/13, 38.5%) tests were preserved in some patients with BVP (Appendix A). As in patients with UVH, the relationships between OVAR and VEMP were assessed in patients with BVP. BICs and MOCs were not significantly different between the normal and abnormal N-amplitude cVEMP groups, or between the normal and abnormal n10–p15 amplitude oVEMP groups.

## 4. Discussion

In this study, we analyzed the BICs and MOCs of healthy volunteers, patients with UVH, and patients with BVP during OVAR stimulation. We also compared OVAR parameters with cVEMP and oVEMP parameters.

OVAR is a highly nausea-inducing stimulus, and it is difficult to test its effects over a wide range of conditions within a single individual. Therefore, we performed OVAR tests with constant velocity rotation of 60°/s and a tilt angle of 20°, based on previous studies conducted by other groups in which responses were clearly documented with minimal autonomic symptoms [11,12,13]. We analyzed BIC and MOC values for horizontal eye movements alone, because the responses to vertical and torsional movements were insufficient to allow comparison of differences among groups [12,14].

In healthy volunteers, the BICs were symmetric but had opposite signs and varied depending on the direction of OVAR. In patients with UVH, the sign of BICs in response to affected side rotation was reversed, but that of BICs to the unaffected side rotation was unchanged. Moreover, the absolute BIC values were significantly lower than in normal volunteers (Figure 2B). This difference mainly arose due to a reduction of BIC values during affected side rotation: The absolute BIC value was reduced with OVAR to the affected side compared with the unaffected side (CW in right UVH patients and CCW in left UVH patients).

Importantly, the BICs during affected side rotation were normal, but with an opposite sign in previous studies [11,12]. However, the absolute BIC values were significantly reduced compared with healthy volunteers in this study. Previous studies included patients who had undergone acoustic tumor resection or labyrinthectomy. Although the participants in the present study showed varying degrees of UVH on caloric tests, none of them had undergone surgical ablation. Our participants did not show BICs for the affected side on OVAR stimulation that were opposite to normal, possibly due to residual otolith function.

There were no differences in MOCs between healthy volunteers and patients with UVH, consistent with previous findings. Moreover, MOCs did not differ even among patients with BVP, which was a notable finding and has not been reported previously. MOCs represent a distinct parameter associated with OVAR stimulation, but considering the lack of effect on MOC even in profound BVP, MOCs may not originate from the otolith organ. One speculation to this observation would be the role of another sensory input responsible for producing MOC. Residual sensory input during OVAR in a dark environment, which is unaffected in patients with profound BVP, presumably arises from somatosensory or cervical muscles. Alternatively, MOCs may be temporarily attenuated only when static defects are present, such as spontaneous nystagmus or ocular torsion. OVAR studies in patients of acute stage with spontaneous nystagmus may clarify this speculation.

The OVAR test has been used to evaluate otolith function in patients with unilateral vestibular damage [15], BPPV [16], and acoustic neuroma [17]. Although OVAR studies demonstrated otolith dysfunction in various vestibulopathies, we found no correlation between OVAR and VEMP parameters in this study. In a study that tested interactions of signals arising from the semicircular canal and otolith organs, VOR time constants during step velocity rotation were reportedly shorter for OVAR than for earth vertical axis rotation, which indicated reduced effectiveness of the velocity storage system. In contrast, with very-low-frequency sinusoidal stimulation, the phase lead of the VOR was smaller during OVAR than earth vertical axis rotation, suggesting enhanced efficacy of the velocity storage system during OVAR [18]. Recording eye movements during OVAR is complicated because the VOR includes the semicircular canal–ocular reflex, otolith–ocular reflex, and semicircular canal–otolith interaction. In contrast, VEMP tests record the inhibitory responses of cervical muscles and inferior oblique excitatory responses to sound/vibration stimulation of the otolith organ.

To see if the OVAR is related to caloric test, the sum of peak slow component velocities of the warm and cold stimulation was correlated with BIC. In analysis assuming all the lesion sides in the right ear, caloric responses of the affected right ear were not correlated with BIC of OVAR to clockwise rotation (Figure 6B), and those of the healthy left ear were correlated with BIC of OVAR to counter-clockwise rotation (Figure 6C). BIC was correlated with caloric responses with statistical significance, especially in healthy side and combined analysis (Figure 6A,C). Though OVAR was not correlated with VEMP, it was correlated with caloric responses.

However, the absence of correlation between parameters of OVAR and VEMPs was somewhat unexpected. We speculated the dissociation as follows. OVAR stimulates both the semicircular canal and otolith organs and analyzes responses after disappearance of canal response. However, the response is very weak, showing around 3 deg/s. Besides, the intensity is further reduced when the vestibular function is deteriorated, and it is often difficult to distinguish the difference only when looking into the results of an individual patient. On the other hand, the primary sensors that detect sound and vibration used in VEMP are cochlear hair cells, and a relatively stronger stimulus is necessary to obtain a recordable response from otolith organs. Since both tests may detect consistent but very small responses to rotation or sound/vibration, it can be more difficult to obtain meaningful results if artifacts or noise are contaminated, such as difficult eye opening, blinking, or muscular atrophy due to senile change.

Secondly, otolith organs detect changes in linear motion and gravity, but they are very low-sensitive sensors compared to the semicircular canal. The ampulla of semicircular canal is a very sensitive sensor detecting head rotation. In a previous experimental model, it showed that it can detect angular displacement of the cupula of around 1.2 × 10^−3^ degree of arc for a cat whose lowest neural threshold in a single afferent unit has been reported as around 2 deg/s [19]. The value has been modified to 5.2 × 10^−3^ degree using the data of Curthoys et al. [20]. Compared to semicircular canals as a highly sensitive sensor, otolith organ is a very primitive sensor from the view point of change detector. In a study to detect the orientation to the vertical during water immersion, subjects were immersed in water and then rotated in a tucked position on a rod through 3, 4, or 5 revolutions. Rotation was terminated with the head in one of four positions: upright, inclined forward, down, or back. Upon termination of rotation subjects, it was directed to point in the up direction. There were errors in direction of pointing of as much as 180 degrees. Errors were greatest with the head down or back and least with the head up or forward [21].

In addition to these characteristics of each test and otolith organs (weak test responses of VEMP and OVAR, and very low sensitive sensor of otolith organ), completely different types of stimulation (sound/vibration vs. rotation) and recording responses (myogenic potentials vs. eye movements) can explain this mismatch in the interpretation of OVAR and VEMP.

Evaluation of otolith function has broadened the understanding of many vestibular disorders. VEMP test has been establishing itself as a critical method to evaluate the disease extent of diverse vestibulopathies and for the functional documentation of 3rd mobile window, such as superior canal dehiscence syndrome. Interpretation OVAR responses can expand the understanding of vestibular ocular reflex from caloric test and earth vertical axis rotation test and also can be an important tool to investigate the effects of the canal–otolith interaction and semicircular canal–ocular reflex.

Both VEMP and OVAR tests can be used to evaluate otolith organs and the integrity of the primary vestibular afferent, but we should consider the discrepancy during interpretation, and the characteristics and limitation of each test.

## 5. Conclusions

In patients with UVH, the sign of BIC values was reversed during affected side rotation relative to healthy volunteers. Moreover, the absolute BIC values were reduced in patients with UVH. In patients with BVP, the BICs were markedly lower. MOCs were not reduced in patients with UVH or BVP. There were no statistically significant correlations between VEMP and OVAR parameters.

## Figures and Tables

**Figure 1 jcm-10-00756-f001:**
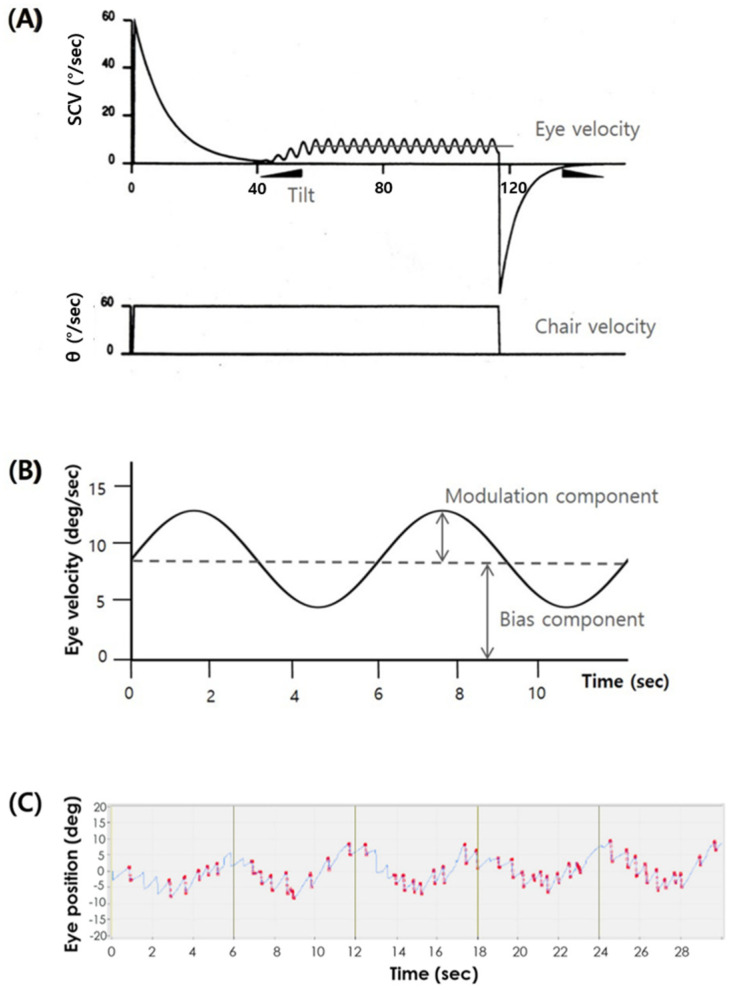
Concept of vestibulo-ocular reflex induced by OVAR and raw tracing during OVAR stimulation. (**A**) During the OVAR test, the VOR generated by the horizontal semicircular canal decreases exponentially, but the VOR generated by the otolith organ is preserved as the direction of gravity changes. (**B**) Ocular slow phase velocities are plotted, such that the center of offset is regarded as the bias component and the amplitude of the sine wave is regarded as the modulation component. (**C**) Raw data of eye movements during OVAR. Horizontal eye position data are digitally differentiated and the fast components of nystagmus identified and removed (area to be deleted in red). Vertical lines every 6 seconds indicate nose down position. Calculated BIC and MOC are 7.55 deg/sec and 3.53 deg, respectively, in this trace.

**Figure 2 jcm-10-00756-f002:**
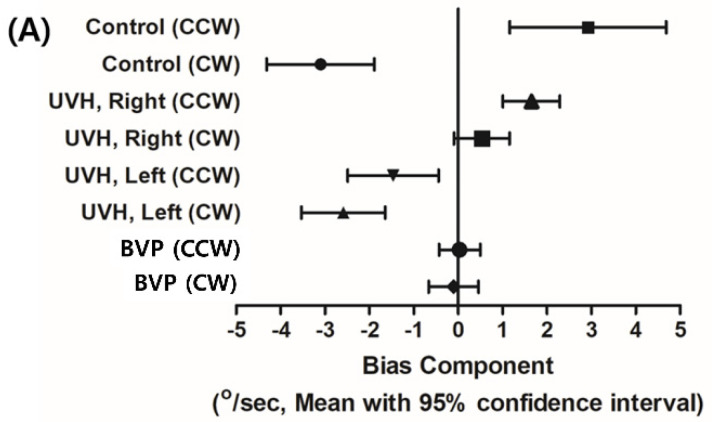
BICs of OVAR in normal volunteers, patients with UVH, and patients with BVP. (**A**) BICs in patients with UVH were lateralized to the unaffected side rotation (CCW in right UVH patients and CW in left UVH patients). (**B**) When all affected ears were assumed to be right ears, the mean BIC values during affected side rotation (CW in right UVH patients) and unaffected side rotation (CCW in right UVH patients) were similar but with opposite signs, indicative of directional preponderance. (**C**) The sums of absolute BIC values were significantly different among groups (* *p* < 0.017, ** *p* < 0.001, Bonferroni correction).

**Figure 3 jcm-10-00756-f003:**
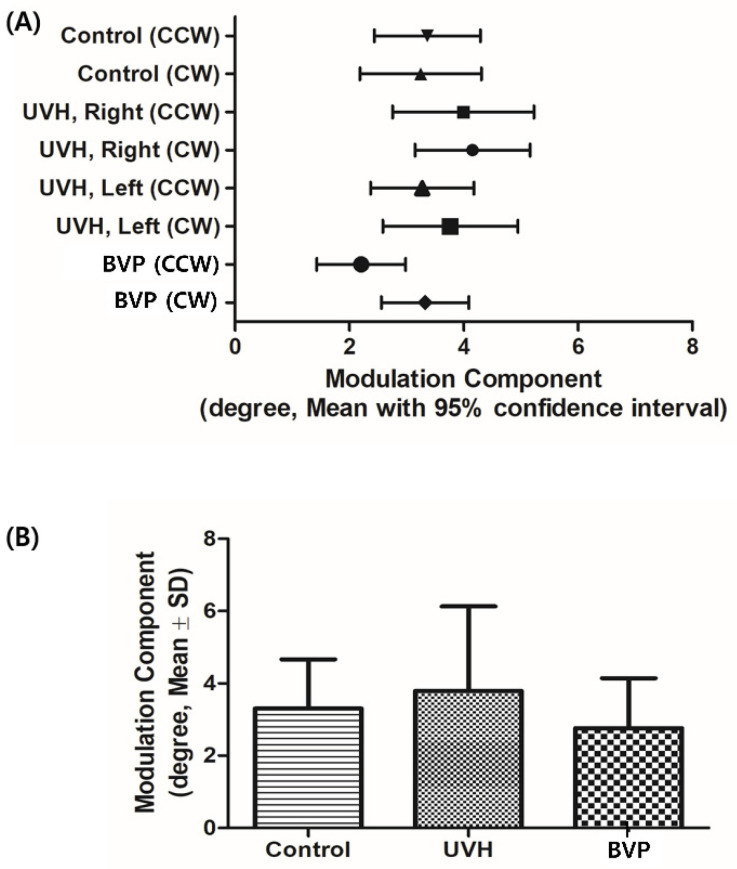
MOCs of OVAR in normal volunteers, patients with UVH, and patients with BVP. (**A**) MOCs show no significant difference between groups and do not change depending on rotation direction. (**B**) The average value of MOCs did not differ significantly among groups.

**Figure 4 jcm-10-00756-f004:**
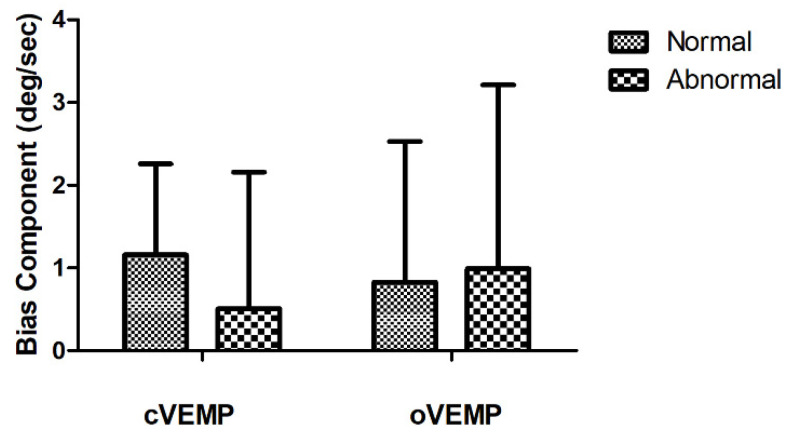
Comparison of BIC according to VEMP results in patients with UVH. There is no significant difference in BIC according to VEMP results.

**Figure 5 jcm-10-00756-f005:**
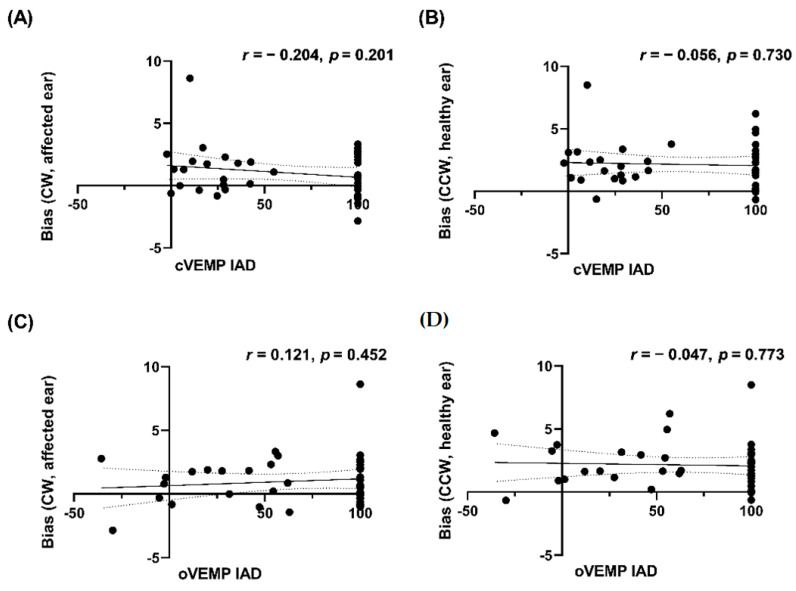
Pearson correlation analyses between VEMP and OVAR parameters in patients with UVH. (**A**,**B**) No significant correlations were evident between IADs of cVEMP and BICs with CW/CCW rotation of OVAR. (**C**,**D**) No significant correlations were evident between IADs of oVEMP and BICs with CW/CCW OVAR.

**Figure 6 jcm-10-00756-f006:**
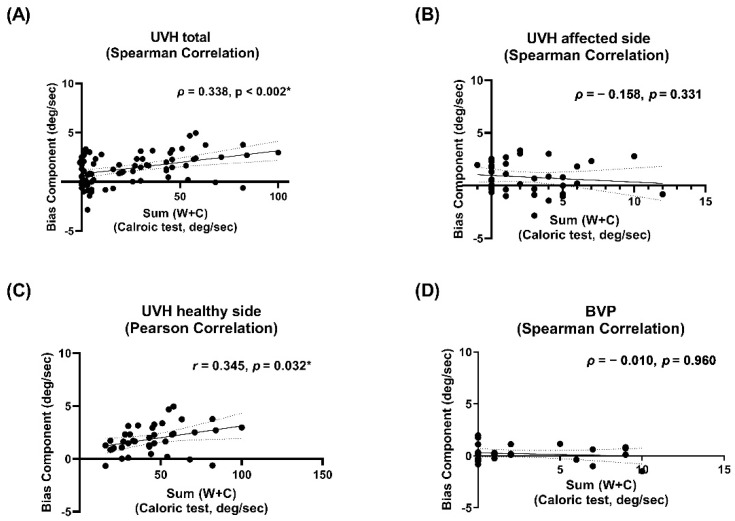
Correlation analyses between VEMP and caloric test in patients with UVH and BVP. Caloric responses show statistically significant correlation (*) with BIC in patients with UVH (**A**), especially in healthy side comparison (**C**). There is no significant correlation in patients with UVH in affected side comparison (**B**) and BVP (**D**).

## Data Availability

The data in this paper can be found in Appendix A.

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
