# Peer review of "Parameters of Off-Vertical Axis Rotation in Unilateral and Bilateral Vestibulopathy and Their Correlation with Vestibular Evoked Myogenic Potentials"

_jcm, 2021, doi:10.3390/jcm10040756_

Round 1

Reviewer 1 Report

Dear authors,

This is an interesting paper with a somewhat expected outcome of the absence of clinical measurements derived directly from the VOR representing otolith information and the vemps. But this absent correlation is by itself necessary to be reported.

There are a few methodological/reporting issues however.

Given the Barany Society paper on Bilateral Vestibulopathy (BVP) (Strupp et al 2017),it is needed to use these criteria for your BVH group, and rename them then also BVP in stead of BVH. The criteria you use mention and in stead of or. Please work this out and use the same criteria as the Barany society. For example you state <= 12 deg/s while Barany is < 6 each side which makes <12 sum, not <=.

Please plot a real trace, and not an ideal model trace like in figure 1. 

Regarding the Vemps:

  • Methods
    • VEMPs: Please mention the stimulus intensity for the c- and oVEMP. Report in decibel Force Level (dB FL) for the oVEMP if possible. 
    • VEMPs: Describe the used electrode configurations for both c- and oVEMP. 
    • Describe how the muscle tension was monitored during the cVEMP test. The muscle tension should be high enough when trying to evoke cVEMPs. Rosengren (2015) wrote a very important paper about this (Rosengren SM. Effects of muscle contraction on cervical vestibular evoked myogenic potentials in normal subjects. Clin Neurophysiol. 2015 Nov;126(11):2198-206. doi: 10.1016/j.clinph.2014.12.027)
    • Line 117: Air conduction cVEMPS: delivered with insert ear phones or headphones?
    • Line 120: Normalized amplitude: Please describe how this parameter was calculated. Depending on how this was calculated, the 'normal range' of  the normalized cVEMP amplitude should be checked as these are high values (2,26-9,78) for a corrected amplitude. If it is normalised for mean tonic activity of the SCM muscle, why is the normalised value than in micro volt. 
    • Line 121: p13-n23 amplitude is a parameter of the cVEMP test but not of oVEMP. For oVEMP, the n10-p15 amplitude should be reported. 
  • Results
    • Lines 216-224: The classification of 'normal' vs 'abnormal' c- and oVEMPs should be included in the Methods. 
    • Line 230: typo: IAC should be IAD  

General use of mean and errors. Please use either Mean ± standard error or Mean (SD), but not Mean (±SD). Please use uniform rounding off rules.

I like figures 2 and 3, however it is essential to plot the means ± 95% confidence intervals in these type of graphs where you compare groups. The use of SD does not make sense to compare groups.

Lines 313 and further on: I think that you should elaborate further on why both techniques report different things. Also, please recommend the use of each method for specific purposes. They seem to be complementary, but what do you exactly learn then from them.

I would rather go for the use of cVEMPS solely for the identification of superior canal dehiscence and oVEMP as a tool to have an idea about the general responsiveness of the utricle, but if you have the OVAR, I think that this method is much closer to the caloric test than the vemps. Perhaps you can make a correlation between caloric test results and VOR gains and the OVAR measurements and see if they are more correlated. Particularly since we know that VEMP parameters don't correlate with gains of VOR tests, neither with caloric outcomes. Please refer to this analogy. One can expect that when one side is down, it may be the entire labyrinth rather than only the horizontal canal only.

Good luck with the adaptations.

Author Response

Thank you for sharing your time and thoughts to improve our manuscript.

Reviewer 2 Report

This paper is written about the results of VEMP and OVAR of UVH and BVH patients.

In method part, the authors should write the method of recording and analysis of eye movement. And the authors should write how to desaccade.

In result part, the authors should show typical waveform of eye position and velocity in a patients.

I have not seen the OVAR data of which MOC was higher than BIC. But in this article, the value of MOC was higher than the value of BIC. The authors should show such a raw data, i.e., waveform of eye position and velocity. 

Author Response

(The authors gave the same response as above.)

Round 2

Reviewer 2 Report

The authors responded to my requests.